# ViSCNOVAS: A Novel Classification System for Hyaluronic Acid-Based Gels in Orthobiologic Products and Regenerative Medicine

**DOI:** 10.3390/gels10080510

**Published:** 2024-08-02

**Authors:** Fábio Ramos Costa, Luyddy Pires, Rubens Andrade Martins, Bruno Ramos Costa, Gabriel Silva Santos, José Fábio Lana

**Affiliations:** 1Department of Orthopedics, FC Sports Traumatology, Salvador 40296-210, BA, Brazil; fabiocosta123@uol.com.br; 2Department of Orthopedics, Brazilian Institute of Regenerative Medicine (BIRM), Indaiatuba 13334-170, SP, Brazil; luyddypires@gmail.com (L.P.); josefabiolana@gmail.com (J.F.L.); 3Regenerative Medicine, Orthoregen International Course, Indaiatuba 13334-170, SP, Brazil; 4Medical School, Tiradentes University Center, Maceió 57038-000, AL, Brazil; rubensdeandrade@hotmail.com; 5Medical School, Zarns College, Salvador 41720-200, BA, Brazil; fabiocosta7113@gmail.com; 6Clinical Research, Anna Vitória Lana Institute (IAVL), Indaiatuba 13334-170, SP, Brazil; 7Medical School, Max Planck University Center (UniMAX), Indaiatuba 13343-060, SP, Brazil; 8Medical School, Jaguariúna University Center (UniFAJ), Jaguariúna 13911-094, SP, Brazil

**Keywords:** hyaluronic acid, orthobiologics, classification, regenerative medicine, clinical applications

## Abstract

Hyaluronic acid (HA), a naturally occurring polysaccharide, holds immense potential in regenerative medicine due to its diverse biological functions and clinical applications, particularly in gel formulations. This paper presents a comprehensive exploration of HA, encompassing its origins, molecular characteristics, and therapeutic roles in gel-based interventions. Initially identified in bovine vitreous humor, HA has since been found in various tissues and fluids across vertebrate organisms and bacterial sources, exhibiting consistent physicochemical properties. The synthesis of HA by diverse cell types underscores its integral role in the extracellular matrix and its relevance to tissue homeostasis and repair. Clinical applications of HA, particularly in addressing musculoskeletal ailments such as osteoarthritis, are examined, highlighting its efficacy and safety in promoting tissue regeneration and pain relief. Building upon this foundation, a novel classification system for HA-based interventions is proposed, aiming to standardize treatment protocols and optimize patient outcomes. The ViSCNOVAS classification system refers to viscosity, storage, chain, number, origin, volume, amount, and size. This classification is specifically designed for HA-based orthobiologic products used in regenerative medicine, including orthopedics, sports medicine, aesthetics, cosmetic dermatology, and wound healing. It aims to provide clinicians with a structured framework for personalized treatment strategies. Future directions in HA research are also discussed, emphasizing the need for further validation and refinement of the proposed classification system to advance the field of regenerative medicine. Overall, this manuscript elucidates the biological functions of hyaluronic acid and its potential in clinical practice while advocating for standardization to enhance patient care in various regenerative applications.

## 1. Introduction

Hyaluronic acid (HA), often referred to as hyaluronan, is a naturally occurring biological compound found in numerous tissues and fluids within the body [1]. Notably, HA possesses unique gel-forming properties, making it a crucial component in various medical and cosmetic applications. These properties enable HA to form hydrogels that exhibit excellent biocompatibility, viscoelasticity, and water retention capabilities, which are essential for its role in regenerative medicine.. It can be obtained from various sources, including synthetic production, biofermentation and extraction from rooster combs, bovine vitreous humor, and umbilical cords [2].

HA was initially identified as glycosaminoglycan (GAG) in 1934 by Karl Meyer and John Palmer from bovine vitreous humor. The term “hyaluronic acid” is derived from “hyaloid,” meaning vitreous, and “uronic acid” [1]. Later on, HA was detected in various organs and tissues, including the skin, joints, and the human umbilical cord, among others. Researchers also found that HA could be synthesized by numerous bacterial species such as *Escherichia coli*, *Bacillus subtilis*, and *Streptococcus zooepidemicus* through fermentation [3]. Conveniently, the molecular arrangement and physicochemical characteristics of HA remain consistent across both vertebrate organisms and bacterial sources [1]. Moreover, it has been observed that HA is synthesized by diverse cell types throughout various phases of the cellular life cycle, serving as a fundamental constituent of the extracellular matrix (ECM) [4]. HA has been under consideration for clinical applications due to its ability address painful musculoskeletal ailments, especially in terms of chronic degenerative conditions such as osteoarthritis (OA) [5,6]. For decades, HA has been employed in the treatment of various health conditions, owing to its distinctive physicochemical attributes and biological functions. Notably, HA establishes robust interactions with both cells and the ECM [4,7].

HA gelation can be achieved through physical, chemical, or enzymatic processes. Physical gelation involves non-covalent interactions, while chemical gelation utilizes crosslinking agents to create covalent bonds, enhancing stability and mechanical strength [8]. Enzymatic gelation requires enzymes like transglutaminase for controlled gel formation [9]. The key properties of HA gels include the following: viscoelasticity (balances viscous and elastic properties, providing cushioning and stress absorption); biocompatibility (well-tolerated by the body, minimizing immune responses); hydrophilicity (retains moisture, aiding in tissue hydration and healing); biodegradability (biodegradable, allowing for temporary applications without surgical removal); and modularity (properties can be tailored by adjusting crosslinking, molecular weight, and concentration) [8,9,10].

The aim of this manuscript is to propose a novel classification system for hyaluronic acid-based gels based on the biological functions of hyaluronic acid and its regenerative medicine potential documented in the literature.

## 2. Evolution of Hyaluronic Acid

The contemporary utilization and advantageous effects of HA for various objectives have been continuously investigated in the scientific literature, spanning from fundamental laboratory investigations to comprehensive clinical trials and systematic reviews. Viscosupplementation employing HA and its derivatives has been indispensable in driving substantial advancements in the treatment of orthopedic conditions since its inception in the 1970s, as first proposed by Endre A. Balazs [11]. Subsequently, during the 1980s, novel HA derivatives surfaced as viable strategies for intra-articular (IA) injections, with the aim of reinstating joint homeostasis and shielding against mechanical injury [12].

According to the literature, Synvisc and Hyalgan emerged as the predominant HA products employed in clinical trials, primarily owing to their recognized safety, efficacy, and sustained effects, notwithstanding the requisite for intra-articular injections [11,13]. Hyalgan has notably demonstrated the ability to augment the viability and replication of human chondrocytes when subjected to reactive oxygen species (ROS) [14]. Since then, additional HA derivatives with various characteristics have been introduced for different purposes, as illustrated in Table 1, based on the work published by Migliore and colleagues [15].

Currently, there are numerous HA formulations commercially available [15] and many variables have been thoroughly discussed in the literature [16,17,18]. However, we were unable to find any publications regarding an actual classification proposal. There still seems to be no “gold standard” treatment approach for complex pathologies. We highly appreciate and acknowledge Dr Balazs’ pioneer work on the nomenclature of HA [17] as well as elegant contributions from other researchers [18]. Yet, several studies give more priority to the molecular weight of hyaluronic acid. This led to the subdivision of HA into three categories: low molecular weight (LMW), ranging from 500,000 to 730,000 Daltons (Da); medium molecular weight (MMW), lying in between 800,000 and 2,000,000 Da; and high molecular weight (HMW), with an average of 6,000,000 Da [16].

Indeed, HA molecular weight is a key factor when considering the biological functions elicited in human tissues (Figure 1). For example, the application of LMW HA leads to limited binding, thus resulting in subdued HA biosynthesis. Conversely, MMW HA prompts more robust binding, stimulating a greater number of HA receptors and consequently enhancing the production of endogenous HA. However, it is worth noting that the exceedingly large molecules found in HMW HA products may not always be optimal. These large domains can restrict the availability of free binding sites on cell surfaces, thereby resulting in relatively weaker stimulation of HA biosynthesis [16]. Nevertheless, it is important for physicians to recognize that molecular weight is not the sole determinant, because other variables also play significant roles and exert just as much influence on clinical outcomes, overall.

Therefore, we introduce the “ViSCNOVAS” classification proposal. This acronym refers to significant variables associated with the therapeutic success of interventions with hyaluronic acid.

## 3. ViSCNOVAS Classification

Breaking down the acronym (Figure 2) and the letters in order of appearance, we have the following: “Vi” for viscosity; “S” for storage; “C” for chain; “N” for numbers; “O” for origin (source); “V” for volume; “A” for amount (amount per milliliter); and another ”S”, for size (molecular weight or size of the HA molecule). Table 2 has been created for further details regarding our proposed classification and each of the variables that we considered to be fundamental. Also, we chose this specific arrangement of letters because it alludes to viscosupplementation (visc) and novelty (“novas”, derived from the Latin word “novus”, which means new).

As an example, applying ViSCNOVAS to one type of HA product would generate: Vi(25) S(4) C(M) N(1) O(A) V(3) A(15.4) S(M).

Each component represents the following:Vi(25): Viscosity of 25 pascal-seconds, suitable for applications requiring high resistance to flow.S(4): Storage at 4 degrees Celsius, indicating the optimal temperature for maintaining product stability.C(M): Chain structure is mixed, combining both linear and cross-linked HA.N(1): Number 1 indicates minimal cross-linking, ideal for superficial applications such as fine-line correction.O(B): Origin from bacterial source (biofermentation), which might be relevant for certain clinical decisions or patient preferences.V(3): Volume of 3 mL to be injected per joint, appropriate for moderate joint conditions.A(30): Total amount of 30 mg of HA in the formulation.S(M): Size with medium molecular weight (1–3 MDa), suitable for a balanced efficacy in terms of tissue penetration and duration.

### 3.1. Coding on the Product

The classification code can be printed directly on the product packaging or included in the product insert. Each variable should be represented with its corresponding letter and value, separated by parentheses. For example, a product label might include the classification as follows:

ViSCNOVAS Code: Vi(25) S(4) C(M) N(1) O(A) V(3) A(15.4) S(M)

### 3.2. Digits and Decimal Places

Viscosity (Vi): Usually expressed as a whole number but can include decimals if needed for precision.Storage (S): Whole number representing temperature. Decimals may also be included for fine tuning.Chain (C): Single letter (L, R, M) denoting chain structure.Numbers (N): Single digit (1–5) indicating the degree of cross-linking or complexity.Origin (O): Single letter (A for animal, B for bacterial).Volume (V): Whole number or decimals, depending on the precision needed for the dosage.Amount (A): Total amount of HA per package, typically expressed as a whole number.Size (S): Single letter or abbreviation denoting molecular weight category.

### 3.3. Expanded Examples of ViSCNOVAS Classification

Example: Ostenil^®^

Viscosity (Vi): Vi(20)Storage (S): S(4)Chain (C): C(L)Numbers (N): N(2)Origin (O): O(B)Volume (V): V(2)Amount (A): A(20)Size (S): S(H)

Printed on the Product: Vi(20) S(4) C(L) N(2) O(B) V(2) A(20) S(H)Interpretation: This product has a viscosity of 20 Pa·s, should be stored at 4 °C, has a linear chain structure, a cross-linking degree of 2, is derived from bacterial sources, has a typical injection volume of 2 mL, contains 20 mg in the package, and has a high molecular weight.Products with multiple components: For products with multiple components, each product’s primary component will be classified using the ViSCNOVAS system, and any additional components will be listed separately in the product description. For instance, Ostenil^®^ Plus would be labeled as follows:Primary component (HA):Viscosity (Vi): Vi(40)Storage (S): S(4)Chain (C): C(M)Numbers (N): N(3)Origin (O): O(B)Volume (V): V(2)Amount (A): A(40)Size (S): S(H)Additional component:Mannitol: 10 mg (1.0%)Printed on the product:Vi(40) S(4) C(M) N(3) O(B) V(2) A(40) S(H) | Additional component: Mannitol 10 mg (1.0%)Products with combined molecular weights: For products with multiple molecular weights, the “Size (S)” variable will list each molecular weight category present in the product, separated by a slash (/).Example: Reneha VisPrimary component (HA):

Viscosity (Vi): Vi(15.4)Storage (S): S(4)Chain (C): C(M)Numbers (N): N(3)Origin (O): O(B)Volume (V): V(2)Amount (A): A(22.4)Size (S): S(L/H)

Printed on the product: Vi(15.4) S(4) C(M) N(3) O(B) V(2) A(22.4) S(L/H)

This detailed coding system provides a comprehensive and easily interpretable framework for both clinicians and patients, ensuring informed decisions are made based on the specific attributes of HA products. By implementing this standardized classification on product labels, clinicians can quickly understand the key attributes of HA formulations, facilitating better patient care and optimized treatment strategies.

## 4. Analysis of ViSCNOVAS Classification Variables

### 4.1. Viscosity

Viscosity refers to a fluid’s resistance to deformation when subjected to shear or tensile stress [19]. Fluids lack a fixed shape and therefore cannot resist deformation, resulting in negligible values for their elastic modulus (E) and shear modulus (G) [20]. Instead, fluids possess a distinct property known as viscosity (η). Viscosity can be defined as the measure of a fluid’s resistance to flow and is governed by Newton’s law of viscosity. This law states that viscosity is the ratio of shear stress (τ) to shear rate (γ·), representing the rate at which a fluid flows [20]. Viscosity is typically measured in pascal-seconds (Pa·s), and the formula used is “η = τ/γ·” [20].

The unique chemical and molecular compositions of biofluids influence their viscosity, potentially impacting the movement of particles within the fluid [19]. In simple terms, a greater viscosity corresponds to a thicker consistency in the fluid, while lower viscosity indicates a thinner texture. This is an important variable for hyaluronic acid products since there are different products available in the market. Their individual characteristics may directly impact their viscosity and therefore dictate the potency of their therapeutic effects. Higher viscosity formulations are typically used for deeper injection sites or to provide more structural support, such as joint lubrication [21], while lower viscosity formulations may be used for finer lines or areas requiring more precise contouring, as is the case with aesthetic applications [20].

### 4.2. Storage

Storage is another variable important for the guaranteed viability, efficacy, and safety of HA. A recent observational study reported that adequate handling and proper storage conditions (monitored refrigeration at 4 °C) avoid the risk of contamination of partially used HA gel fillers by fastidious microorganisms, including common bacterial species (*Staphylococcus aureus*, *Streptococcus pyogenes*, and anaerobes) as well as yeasts and molds [22]. Another cross-sectional study [23] regarding the microbiological safety profile of reusing hyaluronic acid fillers revealed that no fungal or bacterial contamination was observed in hyaluronic acid fillers when they were opened and stored at room temperature under non-aseptic conditions. Although the authors suggest that reusing the remaining material in the syringe could be both safe and cost-effective, careful handling and aseptic techniques are imperative, regardless of the procedure. Other authors have reported that storage conditions have a prominent effect on HA degradation when compared to the initial molecular weight of the sample [24], which may therefore affect its viability. HA can lose approximately 9–15% of its original mass at room temperature (20–25 °C) and 5%–10% when kept in the refrigerator (5–7 °C), for a period up to seven months [25].

### 4.3. Chain

In the context of biopolymers like hyaluronic acid, the term “chain” pertains to its molecular configuration. HA exists as a lengthy polymer, constructed from repetitive units of smaller molecules known as monomers, which are interconnected by glycosidic bonds to form an extended chain [26]. These monomers are disaccharide units comprising N-acetylglucosamine (GlcNAc) and glucuronic acid (GlcA), which are arranged in a linear sequence to construct the extended polymer structure of HA [27]. The molecular chain plays a role in its capacity to retain moisture and provide hydration to tissues, along with its interactions with cells and other components within the extracellular matrix [28]. The length of this chain can vary, influencing the characteristics and roles of the HA product [29]. In many HA products, chains can be linear, reticulated (cross-linked), or mixed (linear + cross-linked). Each type of chain structure in HA has its own unique properties and applications, and the choice of structure depends on the desired outcome and intended use in specific medical, cosmetic, or research settings.

Evidence indicates that cross-linking of HA can extend the duration of its effects when administered intra-articularly [30]. Cross-linking employs the chemical modification of HA molecules, creating bonds between them in order to form a reticulated or “cross-linked” structure. This process alters the properties of HA, including its viscosity, elasticity, and degradation rate [31]. Cross-linked HA tends to degrade at a slower rate [32], leading to prolonged retention within the target tissue. This extended duration of action may enhance the efficacy and longevity of intra-articular HA treatments, for example, potentially reducing the frequency of injections needed for symptom management [30].

However, it must be emphasized that, in order to produce reticulated HA, an enzymatic reaction (which is often cytotoxic) is required. Usually, HA is cross-linked with chemicals like 1,4-butanediol diglycidyl ether (BDDE) and poly (ethylene glycol) diglycidyl ether (PEGDE), thus requiring a safety evaluation at the cellular level [33]. An in vitro study [33] involving human epithelial cells (human keratinocyte and human dermal fibroblast cell lines) revealed that in comparison to PEGDE, BDDE is much more harmful. In this study, BDDE significantly decreased cell viability, increased cytotoxicity (elevated production lactate dehydrogenase), altered cell membrane integrity and morphology, and upregulated the expression of pro-inflammatory markers.

Recently, Sciabica et al. [34] highlighted the development and characterization of a novel cross-linked hyaluronic acid utilizing arginine methyl ester as a cross-linking agent. The use of this natural and safe agent not only improved the resistance to enzymatic degradation compared to linear polymers but also demonstrated promising antibacterial efficacy against S. aureus and P. acnes, making it a favorable option for incorporation into cosmetic formulations and skin applications. Moreover, the impact it has on *S. pneumoniae*, coupled with its remarkable tolerance on lung cells, further qualifies this novel product for utilization even in respiratory tract applications.

As a final consideration, it is crucial to acknowledge that the degradation of HA products typically occurs readily, regardless of their rheological characteristics, physicochemical properties, or manufacturing methods [35]. This awareness is essential for clinicians, especially those with brand preferences, to ensure the safety and efficacy of specific treatments.

### 4.4. Numbers

The “Numbers” variable (N) in the ViSCNOVAS classification system refers to the degree of cross-linking in the HA formulation. The degree of cross-linking is a critical factor that affects the mechanical properties and longevity of the HA gel. The scale ranges from 1 to 5, with 1 indicating minimal cross-linking and 5 indicating maximum cross-linking.

N(1): No or minimal cross-linking, suitable for superficial applications like fine-line correction.N(2): Slightly higher cross-linking, appropriate for moderate-depth tissue integration.N(3): Intermediate cross-linking, used for joint lubrication in moderate arthritis.N(4): High cross-linking, suitable for deep tissue integration and sustained release in severe conditions.N(5): Maximum cross-linking, ideal for long-term applications such as chronic arthritis management.

These examples show how ViSCNOVAS guides the selection of HA products tailored to specific clinical needs.

Overall, the higher the number (1 to 5) for the “N” variable in our classification system, the higher the degree of complexity for HA-based products. These insights may contribute to evidence-based practice and facilitate informed decision-making by healthcare providers and clinicians.

### 4.5. Origin

In regard to origin (or source), HA products primarily come from either animal or bacterial sources. Animal-derived HA is typically extracted from rooster combs or other animal tissues rich in hyaluronan, while bacterial fermentation processes produce hyaluronic acid through microbial synthesis [36]. This implies regulatory considerations as these products must then be purified and analyzed, ensuring quality control and safety [36,37]. Socioeconomic and cultural issues also arise as vegan individuals, for example, will opt for non-animal sources, having to resort to biofermentation-derived HA products [38]. Incidentally, bacterial-derived HA is often preferred for medical and cosmetic applications due to its high purity requirements and reduced risk of potential allergenic or immunogenic reactions compared to animal-derived products, such as avian sources (rooster comb), which are commonly utilized in traditional manufacturing processes [39]. Additionally, bacterial fermentation methods may also offer greater control over production parameters, resulting in more consistent product characteristics and batch-to-batch reproducibility.

Cost is another factor that must be considered. According to a previous study [40], HA production using *Streptococcus zooepidemicus* in batch fermentation reaches 2.5 g/L after 24 h. The process involves centrifugation, diafiltration, treatment with activated carbon, and precipitation with isopropanol. The resulting product is suitable for topical formulations, with an estimated production cost of $1115 per kilogram. Alternatively, in a fed-batch culture scenario with a higher titer of 5.0 g/L, the production cost decreases to $946 per kilogram. Despite higher capital and operating costs in these scenarios, the profitability is also higher. This is because HA intended for injectable use commands a higher selling price, which overcompensates for the increased production costs [40].

### 4.6. Volume

“Volume” encompasses pertinent aspects in treatment planning and administration. Firstly, injection volume (in milliliters) is dictated by the specific treatment indication and anatomical site. Larger volumes (at least 3 mL, depending on the anatomical region) are typically used for the management of orthopedic conditions [41], gynecological complications [42], or in aesthetic volumizing procedures [43,44], whereas smaller volumes (≤1.0 mL) are suitable for cosmetic purposes, including fine-line correction or superficial injections [45].

Treatment protocols often prescribe specific volumes of HA to optimize efficacy and safety, with variations based on individual patient factors and treatment goals. Moreover, the volume of HA injected may be linked to product characteristics such as concentration and viscosity, with higher concentrations or viscosities allowing for smaller injection volumes [46]. The duration of the treatment effect may also be influenced by the injection volume, with larger volumes or multiple injections often resulting in longer-lasting outcomes [47].

Importantly, adhering to the recommended injection volumes and techniques is essential for ensuring patient safety and minimizing the risk of adverse events [48,49]. By considering these factors, healthcare providers can make informed decisions when selecting and administering HA-based treatments, ultimately enhancing patient satisfaction and safety. The volume parameter is intended as a guideline for typical injection volumes. Actual volumes should be determined by the treating physician based on individual patient assessments and clinical judgment.

### 4.7. Amount

The “amount” variable within our classification system introduces another crucial perspective to consider when utilizing HA in treatments. Rather than solely focusing on the general aspect of “concentration”, it extends to the total volume or quantity of HA administered during a procedure. Since the concept of concentration is simply defined by “amount per volume”, this variable is quite similar and closely tied to the previous variable, “volume”. Therefore, amount will also vary in accordance with treatment protocols, taking into account factors such as the specific indication, anatomical site, and patient-related variables like tissue laxity and individual objectives [50]. Additionally, the amount of HA required may be influenced by product characteristics beyond concentration, including its rheological properties and ability to integrate with surrounding tissues [20]. It is worth noting that the term “amount” was deliberately chosen instead of “concentration” to ensure proper fluidity in the acronym. This decision aimed to create an acronym that is sonorously pleasing, easy to remember, and logically cohesive.

Prior research, encompassing both human and animal studies within the field of musculoskeletal tissue and orthopedics, suggests that higher doses of HA (total amount in milligrams) yield more favorable outcomes [51,52,53,54]. These studies have reported dosages ranging from 20 mg to 75 mg, administered through varying numbers of injections. Only one recent single-center, retrospective clinical study on 128 patients with knee OA reported that a triple low-dose (30 mg/2 mL) injection of HMW HA is more effective in improving clinical outcome measures than a single high-dose injection (60 mg/2 mL) over a 12-month period [55].

Nevertheless, healthcare providers must still keep the desired duration of effect in mind. Larger amounts may not always lead to extended and beneficial outcomes, depending on individual patient needs and responses to treatment. Ensuring precise administration of the appropriate HA amount is paramount for both patient safety and achieving optimal, natural-looking results. By carefully assessing each treatment area and adhering to recommended application techniques and number of injections, healthcare providers can effectively design HA-based treatments to meet individual patient needs, ultimately enhancing satisfaction and safety outcomes.

### 4.8. Size

The “Size” variable within our classification system refers to HA molecular weight, which plays a pivotal role in determining its physiological attributes and clinical impact. HA molecules exhibit variations in size, ranging from smaller molecules with lower molecular weights to larger ones with greater mass [56,57]. As previously discussed, this diversity affects crucial aspects (Figure 1), which also include viscosity, elasticity, and the ability to traverse biological barriers, consequently shaping their efficacy across diverse medical and cosmetic applications [24,28]. Distinct size categories may align with specific ranges of molecular weights, including low (L) < 500,000 Da; medium low (ML) 500,000 Da to 1 MDa; medium (M) 1 MDa to 3 MDa; medium high (MH) 3 MDa to 4.5 MD; and high (H) > 4.5 MDa HA products. Therefore, due to their size and subsequent receptor binding, each category carries unique characteristics and therapeutic properties, directly influencing factors like tissue penetration, duration of action, and biological activity [58].

Research underscores the profound impact of HA molecule size on its biological functions and therapeutic effectiveness. For instance, high molecular weight HA exhibits anti-inflammatory and immunomodulatory properties, making them more preferable for applications like viscosupplementation in osteoarthritis management and wound healing, as examples. Conversely, low molecular weight HAs may display superior tissue penetration and cellular absorption in comparison, yet their effects might not be as robust. This would therefore render them suitable for dermatological, cosmetic, and aesthetic procedures such as filler treatments aimed at skin care [59,60,61], for example.

A comprehensive grasp of the size variable in HA classification is imperative for tailoring treatments to specific indications and individual patient requirements. By prudently selecting HA products with appropriate molecular weights, healthcare practitioners can optimize treatment outcomes and elevate patient satisfaction across medical and aesthetic interventions.

## 5. Author’s Note

The ViSCNOVAS classification system holds significant potential for advancing clinical practice in various medical fields, including orthopedics, dermatology, and regenerative medicine. Here are several key prospects and possibilities:Personalized treatment strategies:

The classification system allows clinicians to tailor HA-based treatments to individual patient needs by considering specific variables such as viscosity, molecular weight, and cross-linking degree. For example, patients with severe osteoarthritis might benefit from high-viscosity, highly cross-linked HA formulations for sustained joint lubrication and pain relief.

Enhanced decision-making:

By providing a structured framework, ViSCNOVAS may aid clinicians in selecting the most appropriate HA product based on clinical indications. This structured approach reduces uncertainty and enhances the decision-making process, ensuring optimal treatment outcomes.

Improved standardization:

Standardizing HA-based treatments using the ViSCNOVAS system ensures consistency in clinical practice. This standardization could help in comparing outcomes across different studies and clinical settings, facilitating evidence-based practice and improving overall treatment efficacy.

Broad applicability:

The ViSCNOVAS classification is versatile and can be applied across various fields of regenerative medicine. In dermatology, it can guide the selection of dermal fillers for cosmetic procedures, while in orthopedics, it aids in choosing formulations for joint lubrication and cartilage repair.

Educational tool:

The classification system serves as an educational resource for clinicians and researchers, providing a comprehensive understanding of HA products. It helps in training new practitioners and informing patients about the specific attributes and expected outcomes of their treatments.

Facilitating research:

By categorizing HA products based on detailed parameters, ViSCNOVAS supports clinical research by enabling precise comparisons of different formulations. This fosters innovation and development of future HA-based therapies tailored to specific clinical needs.

Regulatory guidance:

The classification system can assist regulatory bodies in evaluating and approving new HA-based orthobiologic products for regenerative medicine. By providing clear criteria for classification, it ensures that new products meet specific standards of quality and efficacy.

## 6. Future Directions

Hyaluronic acid stands as a pivotal orthobiologic compound with wide-ranging applications in regenerative medicine. Its natural occurrence across various tissues and its ability to be sourced from both biological and synthetic origins underscore its versatility in clinical practice. The consistent molecular structure of HA, regardless of its source, highlights its potential for standardized applications.

ViSCNOVAS represents a novel contribution to the field. To the best of our knowledge, there has been no existing classification in the literature regarding the concepts approached in this manuscript. By delineating specific variables related to HA, our classification aims to provide clinicians with a comprehensive framework for understanding and designing optimal treatment strategies.

This classification facilitates better patient care and promotes standardization within the field of regenerative medicine. The significance of standardizing HA-based treatments cannot be overstated. Our proposed classification system not only fills a gap in the literature but also holds promise for improving patient outcomes. ViSCNOVAS extends beyond orthobiologic products and orthopedics, offering applicability in broader areas of regenerative medicine, including wound healing, sports medicine, aesthetics, and cosmetic dermatology, thereby enhancing its utility and relevance. As the field of regenerative medicine continues to advance, standardization becomes increasingly crucial for ensuring the efficacy and safety of treatments.

In conclusion, ViSCNOVAS presents an innovative framework that not only fosters advancements in research, clinical practice, and standardization within regenerative medicine but also heralds promising avenues for optimizing HA-based gel formulations. These developments hold significant potential for enhancing patient care and treatment outcomes, underscoring the evolving landscape of regenerative medicine and its commitment to excellence in therapeutic innovation.

## Figures and Tables

**Figure 1 gels-10-00510-f001:**
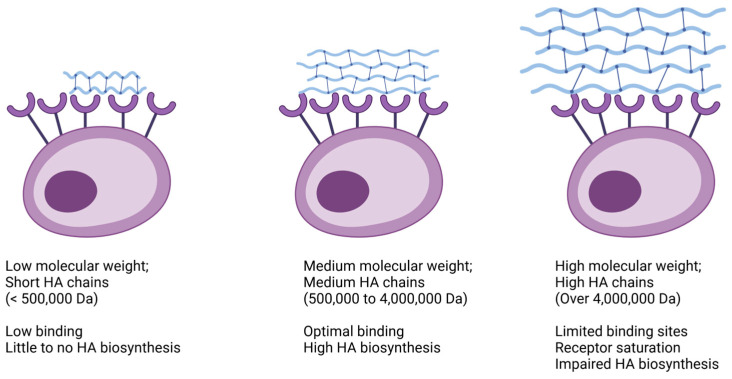
Hyaluronic acid of varying molecular weights and receptor binding.

**Figure 2 gels-10-00510-f002:**
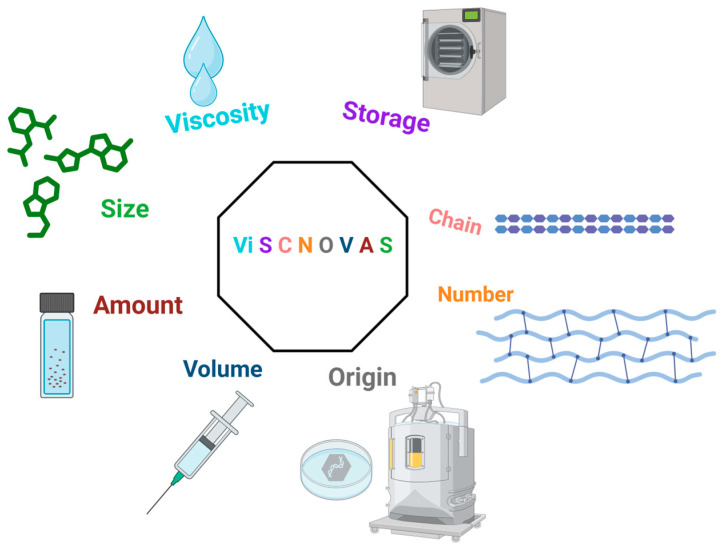
Graphical illustration of the ViSCNOVAS classification system for hyaluronic acid-based gels.

**Table 1 gels-10-00510-t001:** Typical hyaluronic acid formulations utilized in orthopedics.

Source	Product	Concentration (Per mL)	Total Amount in Package	Molecular Weight	Indications
Biofermentation	Ostenil^®^	20 mg (2%)	20 mg	High molecular weight (1–2 MDa)	Used for treating mild to moderate osteoarthritis in the knee.
Biofermentation	OrthoVisc^®^	15 mg (1.5%)	30 mg	High molecular weight (1.1–2.9 MDa)	Provides joint lubrication and pain relief in knee osteoarthritis.
Biofermentation	OrthoVisc^®^ mini	15 mg (1.5%)	15 mg	High molecular weight (1.4 MDa)	Suitable for smaller joints affected by osteoarthritis.
Biofermentation	Synolis^®^ VA	HA: 20 mg (2%) Sorbitol: 40 mg (4%)	60 mg	High molecular weight (2.1 MDa)	Used for joint lubrication and cushioning in osteoarthritis.
Biofermentation	RenehaVis^®^	Low MW HA: 15.4 mg (1.5%) High MW HA: 7 mg (0.7%)	22.4 mg	Low (<1 MDa) and high (2 MDa) MW	Provides dual-action treatment for osteoarthritis, addressing both pain and inflammation.
Biofermentation	MonoVisc^®^	80 mg (8%)	80 mg	High molecular weight (1–2.9 MDa)	Used for single-injection knee osteoarthritis treatment, providing long-lasting relief.
Biofermentation	Ostenil^®^ Plus	HA: 40 mg (4%) Mannitol: 10 mg (1.0%)	50 mg	High molecular weight (1–2 MDa)	Provides extended joint lubrication and pain relief in osteoarthritis.
Rooster Comb	Synvisc^®^	16 mg (80% HMW HA cross-linked; 20% gel)	16 mg	High molecular weight (6 MDa)	Treats severe osteoarthritis with enhanced viscosity for better joint lubrication.
Rooster Comb	Synvisc^®^ One	48 mg (80% HMW HA cross-linked; 20% gel)	48 mg	High molecular weight (6 MDa)	Single-dose treatment for severe osteoarthritis with long-lasting effects.
Biofermentation	Suprahyal Duo	10 mg (1%)	10 mg	Medium molecular weight (1 MDa)	Used for mild osteoarthritis, providing moderate joint lubrication.
Biofermentation	Euflexxa	10 mg (1%)	10 mg	High molecular weight (2.4–3.6 MDa)	Provides effective joint lubrication for knee osteoarthritis, reducing pain and improving function.
Biofermentation	Polireumin	10 mg (1%)	10 mg	Low to medium molecular weight (0.5–1 MDa)	Suitable for mild osteoarthritis and smaller joints, providing joint cushioning and lubrication.
Biofermentation	Reviscon PLUS	16 mg (1.6%)	16 mg	High molecular weight (3 MDa)	Used for treating moderate to severe osteoarthritis, providing sustained joint lubrication and pain relief.
Biofermentation	Biovisc PLUS	20 mg (2%)	20 mg	High molecular weight (3.2 MDa)	Provides effective lubrication and cushioning for knee osteoarthritis, reducing pain and improving mobility.

**Table 2 gels-10-00510-t002:** Description of ViSCNOVAS classification variables.

Letter	Relates to	Description	Possible Values
Vi	Viscosity	Pascal-second (Pa·s) or kg·m^−1^·s^−1^	Numerical values (e.g., 10, 25)
S	Storage	Temperature (degrees Celsius)	Numerical values (e.g., 4, −20)
C	Chain	Linear (L), reticulated (R), or mixed (M)	L, R, or M
N	Numbers	Degree of cross-linking	1, 2, 3, 4, 5
O	Origin	Source of HA	A (animal), B (bacterial)
V	Volume	Typical volume to be injected (mL) per joint or tendon	Numerical values (e.g., 1, 3)
A	Amount	Total amount in package (mg)	Numerical values (e.g., 20, 45)
S	Size	Molecular weight (Da)	Single (e.g., L, M) or combined (e.g., L/H, ML/M)

## Data Availability

The data presented in this study are openly available in article.

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
