# Peer review of "ViSCNOVAS: A Novel Classification System for Hyaluronic Acid-Based Gels in Orthobiologic Products and Regenerative Medicine"

_gels, 2024, doi:10.3390/gels10080510_

Round 1
Reviewer 1 Report
Comments and Suggestions for Authors
This is a most interesting ms and concept and is most suitable for publication. I only question the suitability of it appearing as a paper in "Gels"
The introduction of the paper states " The aim of this report is to describe the biological function of hyaluronic acid ....,... and to propose a new classfication based on specific variables related to this orthobiologic product".
I have searched through the ms to find any reference to the fact that under certain conditions hyaluronic acid forms a gel and how this relates to its medical use to avail.
The authors refer to it as a compoound or a polymer.
The ms reads as a tutorial on the relevant areas of hyaluronic acid. It is not set in the context of its functions as a gel.
I encourage the authors to explore more suitable journals including those in the MPDI portfolio such as Scientia Pharmaceutics, Pharamceuticals etc
As a general comment I suggest it may be helpful to have a short overview of other frameworks for complex medical products.
In summary, good ms but wrong journal it deserves widespread relevant readership
Author Response
Dear reviewer 1, please see attachment.

Reviewer 2 Report
Comments and Suggestions for Authors
In this manuscript, the authors provide a clear and novel classification system ViSCNOVAS for hyaluronic acid-based formulations used as orthobiologic products, providing clinicians with a structured framework for personalised treatment strategies. Since there has been no existing classification in the literature regarding clinical HA formulations, this system has great significance in promoting the standardised applications of HA products in regenerative medicine. This manuscript could be published on Gels after addressing the following comments.
1. Please clarify the formulation catalogues which ViSCNOVAS is used to describe in the title, abstract, introduction, and conclusion. It is implied in the introduction part that this system is only used for orthobiologic products. However, in the abstract and the conclusion part regarding the importance of this system, the catalogue is not emphasised. The applicability means a lot to the significance of this classification system and this manuscript. If the authors tend to apply this system to a wider field, like regenerative medicine including orthopedics, wound healing, cosmetics, etc., more detailed descriptions and examples should be included in the introduction of each letter.
2. Please give examples of the practical application of this system. Although the authors have explained the descriptions of every letter in the acronym, it is still unclear that how does this system work on specific clinical cases. For instance, how would this classification system help with the designing of HA-based treatment strategies for an elderly individual with chronic arthritis or an athlete with meniscus injury?
3. Please reconsider the necessity of inducing “Animal or Bacterial” in the description of “origin”, because animal-derived HA are hardly used in modern regenerative medicine.
4. Please further discuss the classification of “numbers” in the acronym. For instance, what is the corresponding range of the cross-linking degree, polymerisation degree, subunit count and sustainability of numbers 1 to 5? Is there any correlation between a specific indication and a number? If so, please give examples.
5. Please include more details in the introduction of cross-linking methods of HA formulations, which is important for gel study. It is recommended to cite the following references regarding HA-based hydrogels: Sigen A, Jing Lyu, Melissa Johnson, Jack Creagh-Flynn, Dezhong Zhou, Irene Lara-SaÃÅez, Qian Xu, Hongyun Tai, and Wenxin Wang. 'Instant Gelation System as Self-Healable and Printable 3D Cell Culture Bioink Based on Dynamic Covalent Chemistry.' ACS Applied Materials & Interfaces 2020 12 (35), 38918-38924.
6. Please supplement the indications of every product in Table 1.
Author Response
Dear reviewer, please see attachment.

Reviewer 3 Report
Comments and Suggestions for Authors
The authors in their study describe their proposed classification system for hyaluronic acid preparations in terms of various parameters. There are really no descriptions of this kind of systems in the literature, so the novelty of this article is undeniable. However, it has a non-standard format for review.
There are some comments. Table 1 requires links to sources from which information about hyaluronic acid preparations was obtained.
I would recommend that the authors draw up a diagram of the proposed classification, which would clearly show what exactly it offers and what advantages it has.
Perhaps the article would benefit from being more like a review article if the authors included illustrations from the works they cite.
It is necessary to outline in more detail the prospects and possibilities of using this classification in clinical practice.
Author Response
Dear reviewer, please see attachment.

Reviewer 4 Report
Comments and Suggestions for Authors
The article “Hydrogel-based Classification System for Hyaluronic Acid: Introducing the ViSCNOVAS Framework”, tries to devise a method to classify the HA based products, for easy understanding of various parameters mainly by the clinicians. The authors claim that such a classification system would provide a structured framework for clinicians to choose the products for patients’ specific needs. Although the aim of this work is interesting and helps clinicians/common people understand the various parameters of an HA product, there are several limitations in this manuscript. Various major concerns in the manuscript need to be addressed before publication.
1. Although we can understand that the authors try to include various parameters, such inclusion makes the system more complicated to be denoted by a code. Some of the variables are overlapping parameters, that could be denoted by a single variable.
2. Table 1, for Ostenil, 20 mg per ml is 2 % in concentration. The authors are advised to revise this.
3. Table 1, for OrthoVisc, same as above. 1.5% is 15 mg per ml.
4. For most of the products such as Ostenil Plus, MonoVisc, etc, The authors have mentioned the total amount of HA present in one package of the product. Mentioning the total amount under the title "concentration (per mL)" is confusing. The authors should revise it.
5. Figure 1, HA should be shown as chains rather than single molecules. The authors represent the LMW HA as shorter chains and HMW weight as longer chains and so forth.
6. Line 124,125, The authors should expand and explain this example. Furthermore, the authors should also explain, how this classification system will be coded on the product and how many digits/ decimal places are allowed for each variable.
7. Table 2, The possible parameter ranges or the examples of denoting letters should be included as a separate column. For example, for O (Origin), the possible options will be A or B.
8. Table 2, V (Volume), The total volume to be injected per joint or tendon, depends on the pathophysiological conditions of each individual., with the judgment of the physician. How could this be included in a generalized classification system?
9. Section 3.4 Numbers, the authors have made it more complex to understand this variable. It would be better to define a single parameter when devising a classification system and a notation. Including a more general term such as 'Number' and defining it as "may refer to a few concepts", defies the purpose of a classification system that would be easy to understand. The authors could denote a single parameter or concept. Furthermore, the polymerization degree of HA is almost equivalent to that of the molecular weight of HA.
10. Line 256, What are the factors involved in the degree of complexity? For example, if an HA sample has a value of 4 for N, what are all the information, that could be obtained from this variable? The authors should clearly explain, what are all the complexities involved and how they could be numbered from 1 to 5.
11. Section 3.6 Volume, The volume of injection is not a variable for classifying the HA. For example, the same product could be injected more for one patient and lesser for another patient. In this case, how could a product be classified using this variable?
12. The authors should include examples using their classification system so that it will be easy for people to understand. The example should include, how the ViSCNOVAS will be printed or mentioned in a product and how a clinician or a common person could get the necessary information from it. Further, the authors should also show an example, by taking a few existing HA products and classifying them under their classification system.
13. The authors should also provide a provision for a product that has multiple components. For example, Ostenil Plus.
14. Some products use a combination of different molecular weight HA. How could such products be denoted in this classification system? For example, Reneha Vis.
15. Section 3, The values under each variable, their ranges, their corresponding notation, etc., should be clearly explained under each subsection.
Author Response
Dear reviewer, please see attachment.

Round 2
Reviewer 2 Report
Comments and Suggestions for Authors
The concerns have been well addressed by the authors. The importance and comprehensiveness have been largely improved. This manuscript could be accepted by Gels.
Author Response
Dear reviewer 2,
Thank you once again for reviewing our manuscript. We are delighted to know you are satisfied with the corrections.
Best wishes,
The authors.
Reviewer 4 Report
Comments and Suggestions for Authors
The authors have revised the manuscript and all the suggestions has been addressed.
A minor suggestion is that in section 4.4 Numbers, the authors could include the non-crosslinked hyaluronic acid under N(1): No or minimal cross-linking.
The manuscript can be accepted with the abovesaid minor revision.
Author Response
Dear reviewer 4,
Thank you for providing a valuable suggestion. We agree with you. Indeed, N(1) from section 4.4 Numbers encompasses non-crosslinked HA.